# Pharmacoinvasive strategy versus fibrinolytic therapy alone in adults with ST-elevation myocardial infarction: A systematic review and meta-analysis

David R. Soriano-Moreno[1], Kimberly G. Tuco[1], Carolina J. Delgado Flores[2],
Kevin Flores-Lovon[3], Fabricio Ccami-Bernal [3], Renatta Quijano-Escate[4],
Luis Marcos López-Rojas[5], Sergio Goicochea-Lugo [6]*

**1** Unidad de Investigación Clínica y Epidemiológica, Escuela de Medicina, Universidad Peruana Unión, Lima, Peru, **2** Carrera de Farmacia y Bioquímica, Facultad de Ciencias de la Salud, Universidad Científica del Sur, Lima, Peru, **3** Facultad de Medicina, Universidad Nacional de San Agustín de Arequipa, Arequipa, Peru, **4** Sociedad Científica de Estudiantes de Medicina de Ica, Universidad Nacional San Luis Gonzaga, Ica, Peru, **5** Hospital Nacional Alberto Sabogal, EsSalud, Callao, Peru, **6** Unidad de Investigación para la Generación y Síntesis de Evidencias en Salud, Universidad San Ignacio de Loyola, Lima, Peru

\* sgoicochealugo@gmail.com

## Abstract

### Objective

To compare the pharmacoinvasive strategy (PS) versus fibrinolysis alone in adults with ST-segment elevation myocardial infarction.

### Methods

We searched PubMed/MEDLINE, Web of Science, Embase, and Cochrane Library from inception until January 17th, 2025. The review protocol was registered in PROSPERO (CRD42022309130). We included randomized controlled trials (RCTs), assessed risk of bias with the Cochrane Risk of Bias 1.0 tool, and calculated pooled relative risks and mean differences. We used GRADE's minimally contextualized approach to determine the certainty of the evidence.

### Results

We found 7 RCTs (n = 3053). The PS, compared to fibrinolysis alone, may have an important reduction on mortality (3.3% vs 3.9%; −5 per 1000; 95% CI: −16 to +10); and has an important reduction on reinfarction (2.6% vs 4.4%; −19 per 1000; 95% CI: −27 to −6), revascularization (9.0% vs 26.8%; −169 per 1000; 95% CI: −220 to −64), and recurrent ischemia (1.2% vs 5.7%; −46 per 1000; 95% CI: −52 to −27) at 30 days follow up. Similar results were found at longest follow up for the outcomes mentioned. PS probably has an important reduction on mean hospital stay length at longest follow up (−2.47 days; 95% CI: −4.17 to −0.78) and may reduce cardiac failure and

**Data availability statement:** All relevant data are within the manuscript and its Supporting Information files.

**Funding:** The author(s) received no specific funding for this work.

**Competing interests:** The authors have declared that no competing interests exist.

cardiogenic shock at 30 days, but the evidence is very uncertain. PS has trivial or no effect on major bleeding (4.6% vs 5.0%; −4 per 1000; 95% CI: −17 to +13) and may have an important reduction on stroke (0.7% vs 1.3%; −6 per 1000: 95 CI: −10 to +2).

## Conclusions

The PS has an important reduction on reinfarction, revascularization, and recurrent ischemia; probably has an important reduction on mean hospital stay length; and may have an important reduction of mortality and stroke. Also, PS may reduce cardiac failure and cardiogenic shock, but the evidence is very uncertain. The risk of major bleeding did not increase.

## Introduction

ST-segment elevation myocardial infarction (STEMI) is the acute ischemia of the entire thickness of the myocardium and represents 30% of acute coronary syndromes [1]. STEMI is a public health problem in both industrialized and developing countries with an estimated global prevalence of over 3 million cases per year, making it the predominant acute manifestation of coronary heart disease [2,3]. Despite advancements in therapy over the years, mortality rates associated with STEMI remain high, particularly in low-middle-income countries that face resource constraints and limited access to comprehensive healthcare [4,5].

The primary objective in the treatment of patients with STEMI is to achieve prompt reperfusion of the myocardium. For this purpose, strategies with catheter-based procedures, pharmacotherapy, or a combination of both can be used. Percutaneous coronary intervention (PCI) is a catheter-based strategy that is preferred when performed by an experienced team. Ideally, in centers with and without primary PCI capability, the procedure should be completed in less than 60 or 90 minutes, respectively [6]. On the other hand, pharmacological therapy is based on fibrinolytics agents, in which case it is called fibrinolysis, fibrinolytic therapy, or thrombolysis. Fibrinolysis reduce mortality when compared with no fibrinolysis [7]. Nevertheless, when compared to primary PCI, fibrinolytic therapy may increase the risk of nonfatal reinfarction and mortality [8].

The timing of treatment administration is a critical factor in the management of patients with STEMI. In this sense, the clinical practice guidelines (CPGs) recommend providing timely treatment (<120 min) with PCI [6]. However, this is not feasible in health systems where the reference time exceeds the established time or if STEMI networks are not adequately implemented. As a result, fibrinolysis alone remains the most commonly available and feasible reperfusion strategy in these settings. In this context, the pharmacoinvasive strategy - consisting of early fibrinolysis followed by routine coronary angiography and potential PCI within 2–24 hours - emerges as a practical, guideline-endorsed alternative when timely PCI is not feasible [5,8,9].

Importantly, while primary PCI is the gold standard of treatment, it is important to compare the pharmacoinvasive strategy versus fibrinolysis alone because this

reflects a more realistic and context-appropriate clinical decision pathway in resource-limited environments, where the majority of patients do not have access to timely primary PCI [10,11].

Given its potential utility in healthcare systems with limited access to primary PCI, the objective of this systematic review (SR) is to assess the efficacy and safety of pharmacoinvasive strategy compared with fibrinolysis alone in adults with ST-segment elevation myocardial infarction. The outcomes assessed included mortality, reinfarction, recurrent ischemia, need for revascularization, length of hospital stay, stroke, major bleeding, heart failure, and cardiogenic shock.

## Methods

The systematic review protocol has been registered at PROSPERO (CRD42022309130). We followed the statements for the Preferred Reporting Items for Systematic Review and Meta-analysis (PRISMA) (S1 Table).

### Inclusion and exclusion criteria

We included studies that met the following criteria: 1) Study type: randomized controlled trial (RCT) available in full text, 2) Population: adults (18 years or older) with a first or recurrent episode of acute STEMI, 3) Intervention: pharmacoinvasive strategy, defined as fibrinolysis plus PCI performed over a period of time of 2–24 hours from the end of fibrinolysis administration, 4) Comparator: fibrinolysis, defined as pharmacological reperfusion therapy alone using thrombolytic agents such as streptokinase, alteplase, tenecteplase, or others, 5) Outcomes: primary efficacy outcomes such as mortality, quality of life, recurrent myocardial infarction or reinfarction, heart failure, and cardiogenic shock; primary safety outcomes such as bleeding, stroke (hemorrhagic and ischemic), anaphylaxis or other drug adverse events; secondary outcomes such as recurrent ischemia and hospital length of stay [12].

We excluded case-control, cross-sectional studies, case reports, reviews, editorials, letters, and conference abstracts.

### Literature search and study selection

We searched in PubMed/MEDLINE, Web of Science, EMBASE, and Cochrane Library (CENTRAL) from inception to January 17, 2025, without restrictions on language or publication date. In addition, we searched the clinical trial registry ClinicalTrials.gov and in the citation list of eligible studies. The full search strategy for each database is available in S2 Table.

Duplicates were removed using the EndNote version X8 and study selection was managed with an online software (Rayyan QCRI, Qatar Computing Research Institute). The authors (DRSM, KFL, KGT, CDF, RQE, FCB, and SGL) independently screened titles and abstracts to identify potentially relevant studies for inclusion. Full text studies were independently assessed by the authors (CDF, DRSM, KGT, KFL, and FCB). Disagreements were resolved by a third independent author (SGL), acting as a adjudicator.

### Data extraction

The authors (DRSM, KGT, KFL, and FCB) independently extracted the following information from the included studies: 1) Study characteristics: first author, name of the trial, country of origin and year of publication; 2) Study population: number of participants, age, and sex distribution; 3) Fibrinolysis: fibrinolytic agents, time from symptom onset to fibrinolysis (pre-hospital or in-hospital) in minutes and incidence of rescue PCI; 4) Pharmacoinvasive strategy: PCI technique, fibrinolytic agent, proportion of patients receiving a stent, the time from symptom onset to fibrinolysis (pre-hospital or in-hospital) in minutes, and time from fibrinolysis to PCI in minutes; 5) Outcomes, 6) Follow-up (days), and 7) Funding. Disagreements were resolved through discussion with a fifth author (CDF).

### Risk of bias

Two authors (KFL, FCB) independently assessed the risk of bias in the studies using the Cochrane Risk of Bias (RoB) 1.0 tool for RCTs [13]. Disagreements were resolved through discussion with a third author (SGL). The tool consists of seven

domains: random sequence generation, allocation concealment, blinding of participants and personnel, blinding of outcome assessors, incomplete outcome data, selective reporting of results, and other sources of bias (imbalance in baseline characteristics). Each domain was judged as having low, high or unclear risk of bias. We considered a study to be at low risk of bias if at least 5 domains were judged as low risk domains, according to criteria applied in previous systematic reviews [14]. However, we also considered the qualitative importance of individual domains, and deficiencies in these domains were considered when interpreting the results.

## Statistical analysis

Review Manager (RevMan) version 5.4.1 was used to calculate risk ratios (RR) for dichotomous outcomes and mean difference (MD) for continuous outcomes, both with 95% confidence intervals (CI). We performed meta-analyses using random-effects models (Inverse Variance and Mantel-Haenszel method) [15]. Statistical heterogeneity was assessed using the $I^2$; values less than 40% were considered as not important heterogeneity [16]. Publication bias could not be statistically assessed due to the small number of studies per outcome. However, no evidence of publication bias was observed based on a qualitative evaluation of small, positive, industry-funded trials. Furthermore, GRADEpro software was used to estimate absolute effects with 95% CI for each outcome.

## Subgroup analyses

We pre-specified subgroup analysis according to risk of bias. We could not perform pre-specified subgroup analysis according to stent types, age, comorbidities, presence of bleeding at fibrinolysis, success of the fibrinolytic therapy alone, and history of myocardial infarction, because the included studies did not evaluate or report outcomes stratified by these subgroups.

## Certainty of the evidence

We used the Grading of Recommendations, Assessment, Development, and Evaluations (GRADE) methodology using a minimally contextualized approach to rate the certainty of the evidence for each outcome as high, moderate, low, or very low, based on study design, imprecision, risk of bias, inconsistency, indirectness, and publication bias [17,18]. With GRADE, RCTs start with high certainty of the evidence. To assess imprecision, we followed the GRADE guidelines 34, which recommend establishing minimally important differences (MID) for each outcome [19]. To define the MIDs, we searched PubMed/MEDLINE for studies reporting these values in the target population; however, none were identified. The search strategy is detailed in S3 Table. Consequently, we determined the MIDs for each outcome based on the results of a trial comparing primary PCI (the most effective intervention for STEMI) with fibrinolysis alone (which served as the comparator group in the present study) [20]. For outcomes not specified in that trial, the MIDs were determined by authors consensus. Then, we downgraded the certainty of evidence by one level if the 95% CI crossed a single MID threshold and rated down two levels if the 95% CI crossed both MID thresholds or when the number of events was small. For risk of bias, we downgraded one level if 50–70% of the studies in the meta-analysis were at low risk of bias and downgraded two levels if less than 50% of the studies in the meta-analysis were at low risk of bias. Regarding heterogeneity, we downgraded one level if $I^2$ ranged from 40% to 80% with inconsistency of the direction of the association and downgraded two levels if $I^2$ exceeded 80% with inconsistency of the direction of the association. Publication bias could not be assessed with graphical or statistical methods due to the small number of studies for each outcome.

We presented the results in Summary of Findings (SoF) tables and followed the GRADE guidance for communicating the results [21]. We used the following informative statements according to the certainty of evidence: direct informative statement for high certainty, "probably" for moderate certainty, "may have" for low certainty, and "may have, but the evidence is very uncertain" for very low certainty.

## Results

### Study selection

We found 10602 articles in the systematic search. After eliminating duplicates, 7589 records were examined by title and abstract, of which 49 were reviewed in full text. Finally, seven RCTs were included [22–28]. We identified three additional reports related to the TRANSFER-AMI trial. The first report evaluated the trial's outcomes at one-year follow up [29], the second report focused on results for outcomes over a longer-term period with a mean follow-up of 7.8 years [30], and the third report compared the intervention effects between male and female participants [31]. No additional studies were found after searching the references of included studies (**Fig 1**). Excluded studies and reasons for exclusion can be found in S4 Table.

### Characteristics of the studies

Seven RCTs were included. Of the 7 studies, 2 were multicenter, conducted in different European countries (France, Italy, Poland, Spain, and Portugal) [24,27]; and the other 5 studies were conducted in a single country: Canada [22,25], Germany [26,28], and Norway [23]. Total sample size was 3053 patients (range: from 163 to 1059), the mean age ranged from 57.6 to 65.0 years, and the prevalence of male in the studies ranged from 20.4% to 85.9%. All the studies reported their results at 30 days follow-up. Also 3 studies reported results at 6 months [22,26,28] and two at 12 months [23,27].

Three studies administered tenecteplase as fibrinolytic agent [22,23,25], two used reteplase [24,28], two used abciximab plus reteplase [26] and one used alteplase [27]. Regarding PCI technique, four studies used stent [24], balloon [28] or non-specific angioplasty [22,27]; and three studies used guidewire with stent (current gold standard) [23,25,26]. In addition, in the fibrinolysis group, the incidence of rescue PCI ranged from 14% to 34.9% (Table 1).

### Risk of bias

Overall, in most of the studies, the domains of the RoB 1.0 tool were rated as low risk of bias. Although, for allocation concealment (5/7 studies had an unclear risk of bias), blinding of participants and personnel (4/7 studies had a high risk of bias); blinding outcome (1/7), selective reporting (1/7), and other bias (1/7) had a high risk of bias; random sequence generation (1/7) had an unclear risk of bias (Fig 2).

### Minimally important differences

The MIDs (expressed in number of cases per 1000 patients) for the outcomes were the following: mortality: -2 and +2; reinfarction: -3 and +3; revascularization: -30 and +30; cardiac failure: -15 and +15; cardiogenic shock: -15 and +15; recurrent ischemia: -15 and +15; hospital stay length: -2 and +2 days; major bleeding: -17 and +17; stroke: -3 and +3.

### Pharmacoinvasive strategy vs fibrinolysis alone

In comparison with fibrinolysis alone, pharmacoinvasive strategy had the following results (Table 2). The forest plots of each meta-analysis can be found in S1 Fig.

**Mortality.** Pharmacoinvasive strategy may have an important reduction on mortality at 30 days (7 RCTs; 5 fewer per 1000; 95% CI: from 16 fewer to 10 more; low certainty of evidence) (Fig 3) and may have an important reduction on mortality at longest follow-up (range: 30 days to 12 months) (7 RCTs; 13 fewer per 1000; 95% CI: from 25 fewer to 6 more; low certainty of evidence) (Fig 4)**.**

**Reinfarction.** Pharmacoinvasive strategy has an important reduction on reinfarction at 30 days (7 RCTs; 19 fewer per 1000; 95% CI: from 27 fewer to 6 fewer; high certainty of evidence) (Fig 5) and has an important reduction on reinfarction at longest follow-up (range: 30 days to 12 months) (7 RCTs; 23 fewer per 1000; 95% CI: from 33 fewer to 9 fewer; high certainty of evidence) (Fig 6)**.**

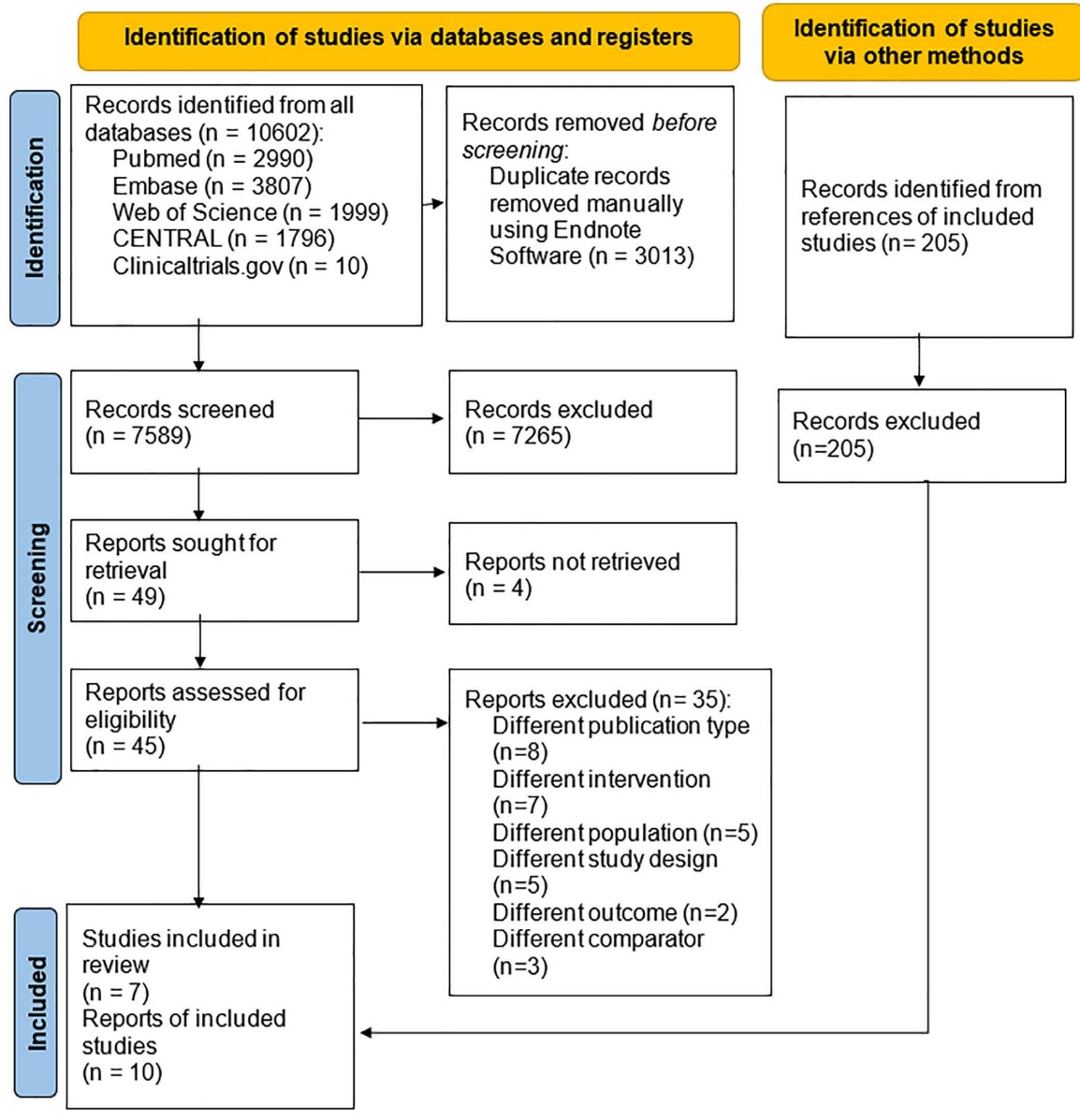

**Fig 1. PRISMA selection flow diagram.**

**Revascularization.** Pharmacoinvasive strategy has an important reduction on revascularization at 30 days (2 RCTs; 169 fewer per 1000; 95% CI: from 220 fewer to 64 fewer; high certainty of evidence) and has an important reduction on revascularization at longest follow-up (range: 30 days to 12 months) (4 RCTs; 140 fewer per 1000; 95% CI: from 159 fewer to 115 fewer; high certainty of evidence).

**Cardiac failure.** Pharmacoinvasive strategy may reduce cardiac failure at 30 days, but the evidence is very uncertain (2 RCTs; 21 fewer per 1000; 95% CI: from 42 fewer to 18 more; very low certainty of evidence).

**Cardiogenic shock.** Pharmacoinvasive strategy may reduce cardiogenic shock at 30 days, but the evidence is very uncertain (3 RCTs; 6 fewer per 1000; 95% CI: from 25 fewer to 40 more; very low certainty of evidence).

**Table 1. Characteristics of the included studies.**

| Authors, trial name, and publication year | Sample size | Time from symptom onset to fibrinolysis (minutes)* | | Time from fibrinolysis to PCI (minutes)* | Patients in the intervention group who underwent PCI | Endpoints | Follow-up (month) | Funding |
|---|---|---|---|---|---|---|---|---|
| | | Intervention | Control | | | | | |
| Bøhmer et al., NORDISTEMI, 2010 | 266 | 117 [80–195] | 126 [80–195] | 163 [137–191] | 115 (86%) | Primary outcomes: Mortality, reinfarction, stroke, new myocardial ischemia, bleeding. Secondary outcomes: Transport-related complications, and infarct size. | 1 and 12 | Scientific Board of the Eastern Norway Regional Health Authority, Hamar, Norway; Ada and Hagbarth Waage's Humanitære og Veldedige Stiftelse, Oslo, Norway; and the Innlandet Hospital Trust, Hamar, Norway |
| Cantor et al., TRANSFER-AMI, 2009 | 1059 | 113 [74–182] | 115 [75–191] | 234 [186–294] | 445 (98%) | Primary outcomes: Mortality, reinfarction, recurrent ischemia, new or worsening congestive heart failure, cardiogenic shock, bleeding. Secondary outcome: Transfusion. | 1 and 6 | Canadian Institutes of Health Research, Roche Canada, and Abbott Vascular Canada. |
| Di Mario et al., CARESS-in-AMI, 2008 | 598 | 165 [115–254] | 161 [120–245] | 135 [96–175] | 245 (96%) | Primary outcomes: Mortality, reinfarction, and refractory myocardial ischemia, bleeding, cerebrovascular events. Secondary outcomes: Length of stay in hospital, revascularization (PCI or coronary artery bypass graft). | 1 | Italian Society of Interventional Cardiology. |
| Armstrong et al., WEST, 2006 | 304 | 130 [75–185] | 113 [74–179] | Not specified | Not specified | Primary outcomes: Mortality, reinfarction, refractory ischemia, congestive heart failure, cardiogenic shock, and major systemic bleeding, and disabling stroke Secondary outcomes: 90- and 180min ST-resolution and the infarct size, intracranial hemorrhage, and major ventricular arrhythmia. | 1 | Hoffman-La Roche and Sanofi-Aventis Canada and also Eli Lilly Canada |
| Thiele et al., 2005 | 164 | 92 [68–179] | 90 [60–150] | Not specified | 82 to 100% | Primary outcomes: Mortality, reinfarction, disabling stroke, or major bleeding. Secondary outcomes: Final infarct size, and ST-segment resolution at 90 min. | 1 and 6 | Lilly Deutschland GmbH, Bad Homburg, Germany and Roche Deutschland GmbH, Grenzach-Wyhlen, Germany. |
| Fernandez-Avilés et al., GRACIA I, 2004 | 499 | 182 (113) | 187 (121) | 1002 (336) | 199 (80%) | Primary outcomes: Mortality, reinfarction, and ischemia-induced revascularization, bleeding, and stroke. Secondary outcomes: Revascularization induced by non-invasive stress tests, readmission due to ischemia. | 1 and 12 | Spanish Ministry of Health, the Spanish Network for Cardiovascular Research (National Health Institute "Carlos III"), the Spanish Society of Cardiology, Guidant, and Lilly. |
| Scheller et al., SIAM III, 2003 | 163 | 192 (132) | 216 (156) | 210 (138) | 163 (100%) | Primary outcomes: Mortality, reinfarction, cardiogenic shock, ischemic events, major bleeding and stroke. Secondary outcomes: Target lesion revascularization CABG, left ventricular ejection fraction. | 1 and 6 | Not reported |

*Data are in mean (standard deviation) or median [interquartile range].

**Fig 2. Risk of bias of the included studies.**

**Recurrent ischemia.** Pharmacoinvasive strategy has an important reduction on recurrent ischemia at 30 days (4 RCTs; 46 fewer per 1000; 95% CI: from 52 fewer to 27 fewer; high certainty of evidence) and has an important reduction on recurrent ischemia at longest follow-up (range: 30 days to 12 months) (5 RCTs; 65 fewer per 1000; 95% CI: from 84 fewer to 22 fewer; high certainty of evidence).

**Hospital stay length.** Pharmacoinvasive strategy probably has an important reduction on mean hospital stay length at longest follow-up (range: 30 days to 12 months) (2 RCTs; MD: 2.47 days fewer; 95% CI: 4.17 fewer to 0.78 fewer; moderate certainty of evidence).

**Safety outcomes.** Pharmacoinvasive strategy has trivial or no effect on major bleeding (7 RCTs; 4 fewer per 1000; 95% CI: from 17 fewer to 13 more; high certainty of evidence) and may have an important reduction on stroke (7 RCTs; 6 fewer per 1000; 95% CI: from 10 fewer to 2 more; low certainty of evidence). No RCT evaluated anaphylaxis and other adverse events.

### Subgroup analyses

We performed subgroup analyses according to the risk of bias to explain the heterogeneity for the outcomes of cardiogenic shock at 30 days ($I^2$: 47%), recurrent ischemia at 30 days ($I^2$: 49%), and recurrent ischemia at the longest follow-up time ($I^2$: 76%). In the case of cardiogenic shock, the study with a low risk of bias showed a favorable effect towards the pharmacoinvasive strategy. On the other hand, studies with a high risk of bias favored fibrinolysis alone; however, the test for differences between subgroups showed no significant differences (p = 0.15). Regarding recurrent ischemia at 30 days and at the longest follow-up, the heterogeneity remained high (S2 Fig).

### Certainty of evidence

The certainty of evidence was high for reinfarction at 30 days, reinfarction at longest follow-up, revascularization at 30 days, revascularization at longest follow-up, recurrent ischemia at 30 days, recurrent ischemia at longest follow-up, and

**Table 2. Summary of findings.**

| Population | Patients with ST elevation myocardial infarction | | | | | |
|---|---|---|---|---|---|---|
| Intervention | Pharmacoinvasive strategy | | | | | |
| Comparator | Fibrinolysis alone | | | | | |
| Outcomes | N° of partici-pants (studies) | Pharmacoinva-sive strategy | Fibrinolysis alone | Relative effect (95% CI) | Absolute effects | Quality of the evi-dence (GRADE) |
| Mortality Follow-up: 30 days | 2946 (7 RCT) | 49/1480 (3.3%) | 57/1466 (3.9%) | RR 0.87 (0.59 to 1.27) | 5 fewer per 1000 (from 16 fewer to 10 more) | ⊕⊕◯◯ Low[a] |
| Mortality Follow-up: longest (30 days to 12 months) | 2925 (7 RCT) | 62/1470 (4.2%) | 77/1455 (5.3%) | RR 0.76 (0.52 to 1.12) | 13 fewer per 1000 (from 25 fewer to 6 more) | ⊕⊕◯◯ Low[a] |
| Reinfarction Follow-up: 30 days | 2946 (7 RCT) | 38/1480 (2.6%) | 65/1466 (4.4%) | RR 0.58 (0.39 to 0.86) | 19 fewer per 1000 (from 27 fewer to 6 fewer) | ⊕⊕⊕⊕ High |
| Reinfarction Follow-up: longest (30 days to 12 months) | 2917 (7 RCT) | 52/1467 (3.5%) | 85/1450 (5.9%) | RR 0.60 (0.43 to 0.84) | 23 fewer per 1000 (from 33 fewer to 9 fewer) | ⊕⊕⊕⊕ High |
| Revascularization Follow-up: 30 days | 760 (2 RCT) | 34/379 (9.0%) | 102/381 (26.8%) | RR 0.37 (0.18 to 0.76) | 169 fewer per 1000 (from 220 fewer to 64 fewer) | ⊕⊕⊕⊕ High |
| Revascularization Follow-up: longest (30 days to 12 months) | 1259 (3 RCT) | 43/627 (6.9%) | 132/632 (20.9%) | RR 0.33 (0.24 to 0.45) | 140 fewer per 1000 (from 159 fewer to 115 fewer) | ⊕⊕⊕⊕ High |
| Cardiac failure Follow-up: 30 days | 1262 (2 RCT) | 31/640 (4.8%) | 44/622 (7.1%) | RR 0.71 (0.40 to 1.25) | 21 fewer per 1000 (from 42 fewer to 18 more) | ⊕◯◯◯ Very low[a, b] |
| Cardiogenic shock Follow-up: 30 days | 1425 (3 RCT) | 30/722 (4.2%) | 28/703 (4.0%) | RR 0.85 (0.36 to 2.00) | 6 fewer per 1000 (from 25 fewer to 40 more) | ⊕◯◯◯ Very low[a, b, c] |
| Recurrent ischemia Follow-up: 30 days | 2084 (4 RCT) | 13/1049 (1.2%) | 59/1035 (5.7%) | RR 0.20 (0.08 to 0.52) | 46 fewer per 1000 (from 52 fewer to 27 fewer) | ⊕⊕⊕⊕ High |
| Recurrent ischemia Follow-up: longest (30 days to 12 months) | 2583 (5 RCT) | 63/1297 (4.9%) | 128/1286 (10.0%) | RR 0.29 (0.13 to 0.63) | 65 fewer per 1000 (from 84 fewer to 22 fewer) | ⊕⊕⊕⊕ High |
| Mean hospital stay Follow-up: longest (30 days to 12 months | 1096 (2 RCT) | Mean range: 7.1 to 7.3 days | Mean range: 9 to 10.5 days | – | MD 2.47 days fewer (4.17 fewer to 0.78 fewer) | ⊕⊕⊕◯ Moderate[d] |
| Major bleeding Follow-up: lon-gest (30 days to 12 months) | 2946 (7 RCT) | 68/1480 (4.6%) | 73/1466 (5.0%) | RR 0.91 (0.66 to 1.26) | 4 fewer per 1000 (from 17 fewer to 13 more) | ⊕⊕⊕⊕ High |
| Stroke Follow-up: longest (30 days to 12 months) | 2946 (7 RCT) | 10/1480 (0.7%) | 19/1466 (1.3%) | RR 0.55 (0.26 to 1.15) | 6 fewer per 1000 (from 10 fewer to 2 more) | ⊕⊕◯◯ Low[a] |
| Quality of life | The studies found did not provide information for this outcome | | | | | |

a. Downgraded two level for very serious imprecision.

b. Downgraded two level for very serious risk of bias.

c. Downgraded one level for serious heterogeneity.

d. Downgraded one level for serious imprecision.

RCT: randomized controlled trial, RR: risk ratio, CI: confidence interval, MD: mean difference.

⊕⊕⊕⊕ High certainty: We are very confident that the true effect lies close to that of the estimate of the effect.

⊕⊕⊕◯ Moderate certainty: We are moderately confident in the effect estimate: The true effect is likely to be close to the estimate of the effect, but there is a possibility that it is substantially different

⊕⊕◯◯ Low certainty: Our confidence in the effect estimate is limited: The true effect may be substantially different from the estimate of the effect.

⊕◯◯◯ Very Low certainty: We have very little confidence in the effect estimate: The true effect is likely to be substantially different from the estimate of effect.

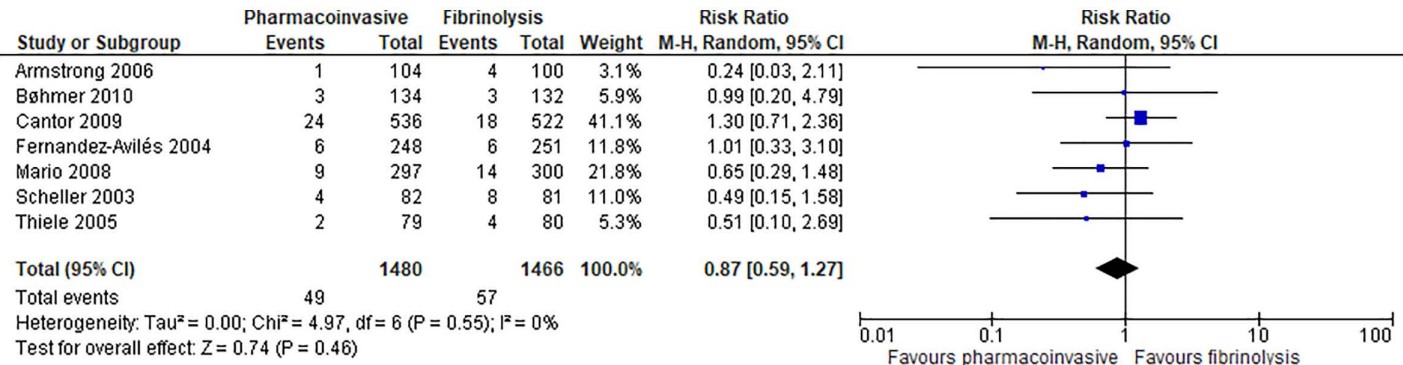

**Fig 3. Forest plot: Mortality (follow-up: 30 days).**

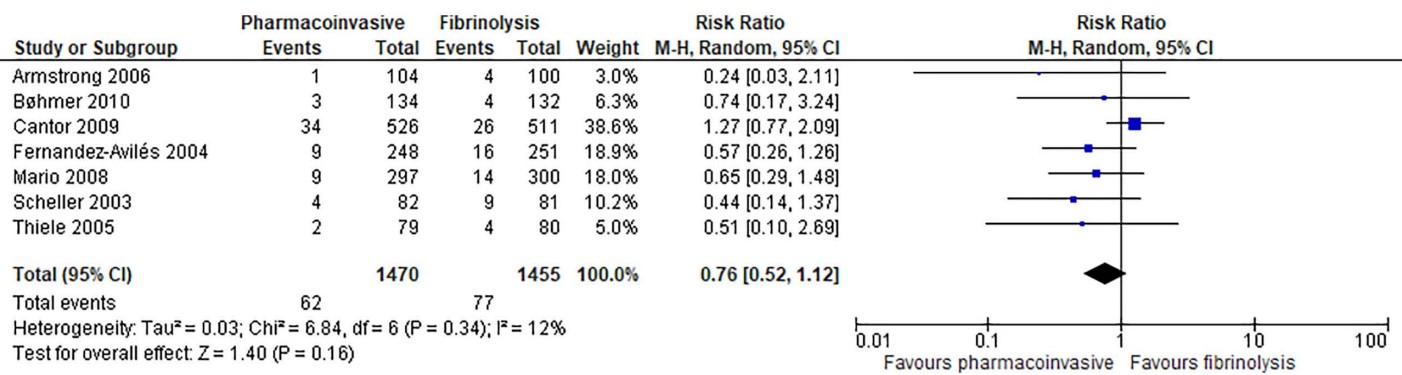

**Fig 4. Forest plot: Mortality (longest follow-up: 30 days to 12 months).**

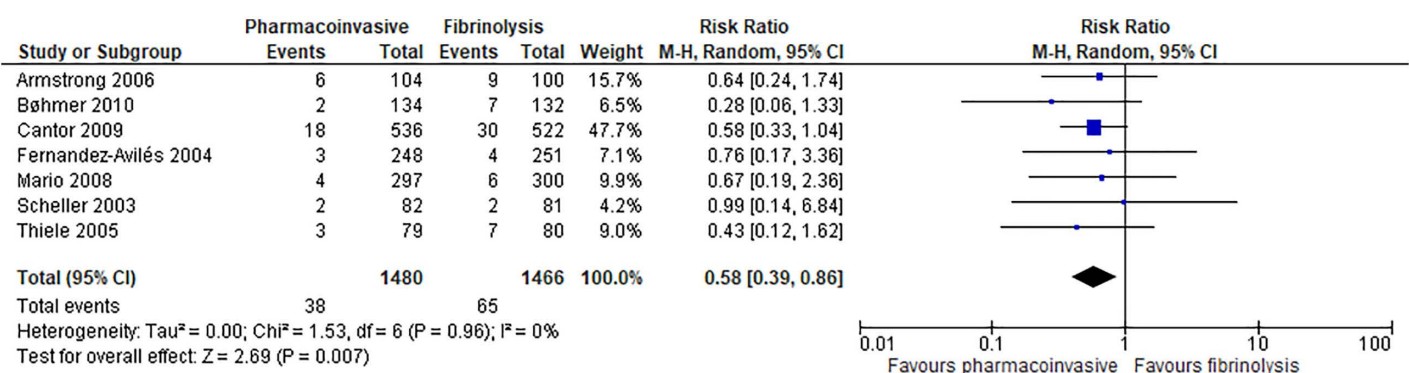

**Fig 5. Forest plot: Reinfarction (follow-up: 30 days).**

major bleeding; was moderate for hospital stay length due to serious imprecision.; was low for mortality (at 30 days and at the longest follow-up), and stroke due to very serious imprecision; and was very low for cardiac failure and cardiogenic shock due to very serious imprecision and very serious risk of bias for cardiac failure, and additionally due to serious heterogeneity (I²: 47%) for cardiogenic shock.

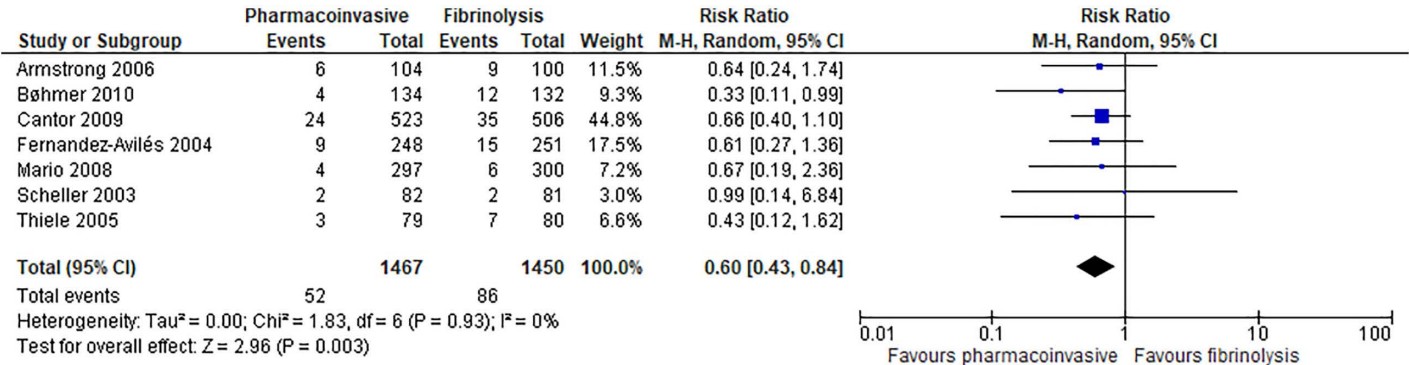

**Fig 6. Forest plot: Reinfarction (longest follow-up: 30 days to 12 months).**

## Sensitivity analysis

In the sensitivity analysis excluding studies at high risk of bias, the results remained consistent with the primary analysis. The benefit of the intervention was more pronounced for outcomes such as mortality, myocardial infarction, and cardiogenic shock, though the overall conclusions were unchanged. For major bleeding, stroke, and recurrent ischemia, effect estimates were comparable to those observed in the main analysis. (S3 Fig).

## Other findings

In the TRANSFER-AMI trial, Bhan et al. compared the effect of the pharmacoinvasive strategy in men and women and found no differences between the two groups in mortality, reinfarction, recurrent ischemia, heart failure, cardiogenic shock, transient ischemic attack/stroke and bleeding at 30 days [31]. In this same trial, Arbel et al. evaluated the effects of long-term (mean follow-up: 7.8 years) pharmacoinvasive strategy and reported no differences in mortality, reinfarction, transient ischemic attack/stroke and heart failure [30].

## Discussion

### Main findings

Our SR included 7 RCTs involving a total of 3053 patients with STEMI that compared the efficacy and safety of pharmacoinvasive strategy versus fibrinolysis therapy alone. Compared to fibrinolysis, pharmacoinvasive strategy may have an important reduction on mortality and has an important reduction on reinfarction, revascularization, and recurrent ischemia. Also, probably has an important reduction on hospital stay length. Additionally, it may reduce cardiac failure and cardiogenic shock, but the evidence is very uncertain. Regarding safety outcomes, pharmacoinvasive strategy has trivial or no effect on major bleeding and may have an important reduction on stroke. None of the studies reported assess quality of life and no other safety outcomes were reported.

### Overall completeness and applicability of evidence

We did a comprehensive literature search to find all possible studies published in scientific journals. The included studies provided sufficient information to carry out our review. However, few RCTs assess cardiac failure, cardiogenic shock, and hospital stay length. Unfortunately, we found no evidence regarding quality-of-life (a pre-specified outcome of interest). The RCTs were representative of clinical practice in several developed countries, including France, Italy, Poland, Spain, Portugal, Canada, Germany and Norway. Unfortunately, low- and middle-income countries account for approximately 80% of the global population, and access to the treatment of choice (primary PCI) is often limited [5]. This limitation is

especially pronounced in rural and underserved areas, where specialized centers are scarce and timely transportation is frequently not feasible. Moreover, findings from studies conducted in high-income countries may not be directly applicable to low-middle income countries due to substantial differences in healthcare infrastructure, availability of trained personnel, emergency response systems, comorbidity burden, and sociodemographic characteristics. These differences can influence both the effectiveness and safety of therapeutic strategies such as the pharmaco-invasive approach.

It should be noted that the number of events in the outcomes assessed by the RCTs could be lower than those found in real clinical scenarios. For instance, while the mortality rate in the intervention and control groups was 3.3% and 3.9%, respectively, the mortality rate from STEMI in other scenarios is typically higher (around 9–10%) [32,33]. Therefore, the benefits of the intervention are likely to be even greater in a real clinical setting. In addition, we found variability according to the type of fibrinolysis and PCI used in the studies. Nonetheless, the results were consistent across the RCTs. The time between fibrinolysis and PCI ranged from 135 minutes (2.25 hours) to 1002 minutes (16.7 hours). However, due to the limited number of studies available, we were unable to perform a meta-regression analysis to determine whether time had any influence on treatment efficacy. Nonetheless, evidence from observational studies suggests that performing PCI as early as possible within the pharmacoinvasive strategy is associated with greater clinical benefit [34]. Another important finding is that, among patients who received fibrinolysis alone, the need for rescue PCI ranged from 14.0% to 34.9% across studies. These findings highlight that a considerable proportion of patients may not achieve adequate reperfusion and ultimately require urgent invasive intervention, underscoring the potential benefits of a pharmacoinvasive strategy.

### Quality of the evidence

Although most domains of the RoB 1.0 tool were judged as low risk in the included studies, several critical aspects should be considered when interpreting our findings. The most frequent limitation was unclear allocation concealment in five of the seven studies, which could lead to selective enrollment of participants, baseline imbalances, and overestimation of the observed treatment effects. In addition, lack of blinding of participants and personnel was observed in four of the seven studies. For subjective outcomes (e.g., recurrent ischemia), this lack of blinding could result in differences in expectations and decision-making by both participants and physicians. However, the critical outcomes in the present study are less prone to subjective influence. With exception of cardiogenic shock and recurrent ischemia at 30 days, as well as at longer follow-up, most of the meta-analyses indicate low levels of heterogeneity. In the subgroup analysis according to the risk of bias, heterogeneity decreased for cardiogenic shock; however, it remained high for recurrent ischemia.

In the certainty of evidence assessment, the imprecision criteria were the most affected, mainly due to the low number of events. This resulted in reduced certainty of evidence for outcomes such as mortality, heart failure, cardiogenic shock, length of hospital stays, and stroke. On the other hand, we found high certainty for seven outcomes, moderate certainty for one outcome, low certainty for three outcomes, and very low certainty for two outcomes.

### Potential bias in the review process

We conducted a comprehensive literature search for RCTs addressing the research question and also screened the reference list of included studies. However, we did not search studies in the grey literature. We excluded observational studies to focus on RCTs, which minimize bias and provide the most reliable evidence on efficacy and safety. In this sense, we acknowledge that this may limit the detection of less frequent adverse events. To minimize any potential biases in our review process, we ensured that at least two authors independently performed selection, extraction, and risk of bias analysis. Furthermore, we specifically looked for evidence on critical and important outcomes for both patients and decision-makers and evaluated the certainty of the evidence using the GRADE approach.

### Comparison with previous reviews

In our background search, we found one SR with network meta-analysis [8] and one SR with meta-analysis [35] that assess a similar review question. The SR with network meta-analysis published in 2020 and the SR with meta-analysis published

in 2013, compared the pharmacoinvasive strategy versus fibrinolysis alone in patients with STEMI and reported similar results as our study; however, one of them included less RCTs (5 RCTs) [35] than our review (7 RCTs) and both SRs did not report the certainty of evidence for the outcomes. Evaluating the certainty of evidence is essential for decision-making, as it enables reliable clinical decisions and reduces harm to patients [36,37]. In addition, our study proposes MIDs to contextualize the importance of the effects taking into account what is clinically relevant for the patient and clinical specialists.

### Implications for practice

Compared to fibrinolysis alone, the pharmacoinvasive strategy reduced mortality, reinfarction, revascularization, recurrent ischemia, hospital stay and stroke with a certainty of evidence ranged from low to high. In addition, it does not increase major bleeding with high certainty of evidence. This findings support the American Heart Association and European Society of Cardiology guidelines, which advocate transferring patients to a center capable of performing PCI after fibrinolysis [6,38]. This is especially relevant in settings where access to PCI within the first few hours of STEMI onset is limited. The results of the present review will be informative for clinicians and stake-holders in their decision-making process, taking into account the clinical impact on significant patient outcomes and hospital resource management. Additionally, we provided MIDs that may be useful for other working groups in their decision-making process.

### Implications for research

While the current evidence suggests the benefits of the pharmacoinvasive strategy in patients with STEMI, further observational studies could be conducted to assess the effectiveness of the pharmacoinvasive strategy in real-world scenarios, especially in developing countries where access to timely primary PCI may be limited. Such studies could provide valuable information on the implementation of this strategy in different healthcare settings and help optimize its use.

### Conclusion

In this systematic review, we included 7 RCTs involving a total of 3053 patients. Compared to fibrinolysis alone, the pharmacoinvasive strategy has an important reduction on reinfarction, revascularization, and recurrent ischemia; probably has an important reduction on mean hospital stay length; and may have an important reduction of mortality and stroke. Also, PS may reduce cardiac failure and cardiogenic shock, but the evidence is very uncertain. In addition, the risk of major bleeding did not increase.

### Supporting information

**S1 Table. Prisma 2020 checklist.**
(DOCX)

**S2 Table. Search strategy.**
(DOCX)

**S3 Table. Search strategy for minimal important differences (MID).**
(DOCX)

**S4 Table. Excluded studies reviewed at full text stage.**
(DOCX)

**S1 Fig. Forest plots.**
(DOCX)

**S2 Fig. Subgroup analyses.**
(DOCX)

**S3 Fig. Sensitivity analysis.**
(DOCX)

## Acknowledgments

Wendy Nieto-Gutiérrez for her methodological support at the beginning of this study.

## Author contributions

**Conceptualization:** Sergio Goicochea-Lugo, David R. Soriano-Moreno, Kimberly G. Tuco, Carolina J. Delgado Flores, Kevin Flores-Lovon, Fabricio Ccami-Bernal, Renatta Quijano-Escate, Luis Marcos López-Rojas.

**Data curation:** Kimberly G. Tuco, Carolina J. Delgado Flores, Kevin Flores-Lovon, Fabricio Ccami-Bernal, Renatta Quijano-Escate.

**Formal analysis:** Sergio Goicochea-Lugo, David R. Soriano-Moreno.

**Investigation:** David R. Soriano-Moreno, Kimberly G. Tuco, Carolina J. Delgado Flores, Kevin Flores-Lovon, Fabricio Ccami-Bernal, Renatta Quijano-Escate, Luis Marcos López-Rojas.

**Methodology:** Sergio Goicochea-Lugo, David R. Soriano-Moreno, Luis Marcos López-Rojas.

**Supervision:** Sergio Goicochea-Lugo, David R. Soriano-Moreno.

**Writing – original draft:** Sergio Goicochea-Lugo, David R. Soriano-Moreno, Kimberly G. Tuco, Carolina J. Delgado Flores, Kevin Flores-Lovon, Fabricio Ccami-Bernal, Renatta Quijano-Escate, Luis Marcos López-Rojas.

**Writing – review & editing:** Sergio Goicochea-Lugo, David R. Soriano-Moreno, Kimberly G. Tuco, Carolina J. Delgado Flores, Kevin Flores-Lovon, Fabricio Ccami-Bernal, Renatta Quijano-Escate, Luis Marcos López-Rojas.

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
