## [Decision Letter · Decision Letter 0]

25 Feb 2025

PONE-D-25-03031Pharmacoinvasive strategy versus fibrinolytic therapy in adults with ST-elevation myocardial infarction: a systematic review and meta-analysisPLOS ONE

Dear Dr. Goicochea-Lugo,

Thank you for submitting your manuscript to PLOS ONE. After careful consideration, we feel that it has merit but does not fully meet PLOS ONE’s publication criteria as it currently stands. Therefore, we invite you to submit a revised version of the manuscript that addresses the points raised during the review process.

We look forward to receiving your revised manuscript.

Kind regards,

Luiz Sérgio Fernandes de Carvalho, PhD, MSc, MD

Academic Editor

PLOS ONE

Journal Requirements:

2. We note that your Data Availability Statement is currently as follows: “All relevant data are within the manuscript and in Supporting Information files.”

3. Please include captions for your Supporting Information files at the end of your manuscript, and update any in-text citations to match accordingly. Please see our Supporting Information guidelines for more information: http://journals.plos.org/plosone/s/supporting-information .

4. As required by our policy on Data Availability, please ensure your manuscript or supplementary information includes the following:

Additional Editor Comments:

As explained below, I listed required and recommended changes for further considering this manuscript.

**Required changes**  primarily stem from Reviewer #1's concerns regarding the misinterpretation and misrepresentation of findings. Specifically, the authors must correct all instances where non-statistically significant differences are described as "possible" or "important" reductions. They should accurately reflect the null hypothesis findings and, if desired, discuss potential type II errors or the need for larger studies. Additionally, the manuscript must address the unclear prevalence statement on page 10 and remove unnecessary uses of "important" for statistically significant differences. **Recommended changes** , from Reviewer #2, involve clarifying methodological choices. While these are not strictly required for technical soundness, they would significantly strengthen the manuscript. These include justifying database selection, addressing the handling of studies with high risk of bias, explaining the chosen risk of bias threshold, clarifying heterogeneity assessment, detailing the application of MID thresholds in GRADE assessment, justifying the two-level downgrading of 30-day mortality, and discussing the impact of variability in time to PCI and gender distribution.

Reviewers' comments:

Reviewer's Responses to Questions

**Comments to the Author**

1. Is the manuscript technically sound, and do the data support the conclusions?

Reviewer #1: Partly

Reviewer #2: Yes

2. Has the statistical analysis been performed appropriately and rigorously? 

Reviewer #1: Yes

Reviewer #2: Yes

3. Have the authors made all data underlying the findings in their manuscript fully available?

Reviewer #1: Yes

Reviewer #2: Yes

4. Is the manuscript presented in an intelligible fashion and written in standard English?

Reviewer #1: Yes

Reviewer #2: Yes

5. Review Comments to the Author

Reviewer #1: PONE-D-25-03031 PLOS ONE Pharmacoinvasive strategy versus fibrinolytic therapy in adults with ST-elevation myocardial infarction: a systematic review and meta-analysis

Review

This is a systematic review and meta-analysis of randomised trials investigating a pharmaco-invasive strategy (fibrinolysis plus early coronary angiography with a view to angioplasty) versus a fibrinolytic therapy (fibrinolysis alone). Seven RCTs were identified, including a total of 3053 patients. No statistically significant difference in mortality was identified, but there was a reduction in reinfarction, unplanned revascularization and angina favouring the pharmaco-invasive strategy.

The systematic review appears to have been rigorously conducted and the statistical analyses sound. The results are however presented and interpreted incorrectly. Even in the abstract the authors state the following:

The PS [pharmaco-invasive strategy], compared to fibrinolysis, may have an important reduction on mortality (-5 per 1000; 95% CI: -16 to +10)

The manuscript is littered with these misrepresentations of findings where the null hypothesis cannot be rejected, described as ‘possible’ reductions. It seems to be a problem with interpretation and misrepresentation rather than a fatal flaw in the analysis, as such I think it can be remedied with what would be quite an extensive re-write. To be suitable for publication in any journal there needs to be a major revision. I would also suggest that some of the important forest plots are put into the main manuscript rather than the supplementary appendix.

Specific comments/questions:

1) On page 10, line 51-53 the authors state:

STEMI is a public health problem in both industrialized and developing countries with a prevalence that can reach up to 40%.

I am not sure what the authors mean here. This suggests that 40% of the population are having a STEMI at any one time. This needs to be corrected.

2) The repeated use of the word important for statistically significant differences between treatment groups is unnecessary and should be changed.

3) Where a statistically significant difference has not been identified stating that there may be an important difference between groups does not mean anything. It might be discussed that a larger patient population would be required to identify a difference and that there is a possibility of type II error. This requires correction in the results and discussion section.

4) Some of the forest plots for the important outcomes should be included in the main manuscript and not just in the supplementary appendix.

Reviewer #2: In this study, the authors conducted a systematic review and meta-analysis comparing the pharmacoinvasive strategy with fibrinolysis in adults with ST-elevation myocardial infarction. The study provides insights into the potential benefits and limitations of pharmacoinvasive management, particularly regarding reinfarction, revascularization, angina, and hospital stay length. My comments and suggestions to the authors are as follows:

1) While the review focuses exclusively on randomized controlled trials, observational studies could provide valuable real-world insights, particularly for safety outcomes such as bleeding and stroke. Could the authors clarify whether observational studies were considered at any stage of the study selection process?

2) The search strategy included PubMed, Web of Science, Embase, and Cochrane Library (CENTRAL), but no justification was provided for excluding other potentially relevant databases such as Scopus, LILACS, or CINAHL. Given that broader searches could reduce publication bias and increase comprehensiveness, could the authors clarify the rationale for limiting the search to these four databases?

3) The manuscript states that publication bias could not be statistically assessed due to the small number of studies (7 RCTs, 3,053 patients). However, the absence of a formal statistical assessment does not preclude a qualitative evaluation. Did the authors conduct any qualitative assessment of publication bias (e.g., reviewing asymmetries in reported outcomes, searching for unpublished trials, or assessing selective reporting)?

4) Figure 2 illustrates that some studies presented a high risk of bias, particularly in blinding of participants and allocation concealment. However, there is no explicit mention of whether studies with high risk of bias were automatically excluded or included in the meta-analysis without restrictions. The Discussion states that most studies had a low risk of bias, but it does not clarify how studies with high risk were handled. Were these studies included in the main analysis? If so, was any sensitivity analysis performed to assess their impact on the overall results?

5) The study considers a trial as 'low risk of bias' if at least 5 out of 7 domains in the Cochrane RoB 1.0 tool are classified as low risk. However, Cochrane guidelines do not recommend a fixed numerical threshold for determining overall risk, as certain domains (e.g., allocation concealment or outcome blinding) can have a disproportionately high impact on validity. Could the authors clarify why this specific threshold was chosen instead of a more qualitative domain-based assessment?

6) The manuscript defines I² <40% as 'not important heterogeneity.' However, according to Cochrane guidelines, an I² between 30-60% can indicate moderate heterogeneity. Could the authors clarify the rationale behind this threshold? Was this classification based on an alternative reference, such as the GRADE Working Group or specific cardiovascular studies? Additionally, some outcomes with I² >40% (e.g., long-term angina, I² = 76%) were not downgraded in certainty. Could the authors clarify why heterogeneity was not consistently considered when assessing the certainty of evidence?

7) The 'Certainty of Evidence' section states that imprecision was assessed according to GRADE guidelines, considering confidence intervals crossing MID thresholds and the number of events. While Table 2 presents the MID values used, it does not specify how they were determined or applied in the downgrading process. Could the authors clarify whether these MID thresholds were based on predefined literature standards, expert consensus, or empirical data from the included studies? Additionally, what specific criteria led to downgrading by one or two levels for each outcome?

8) In Table 2, the certainty of evidence for 30-day mortality was downgraded by two levels due to very serious imprecision. However, the reported risk ratio (RR 0.87, 95% CI: 0.59–1.27) suggests a potentially beneficial effect, though the confidence interval includes 1. Other outcomes with similarly wide confidence intervals were downgraded only one level. Could the authors clarify the rationale for downgrading mortality by two levels instead of one? Additionally, the Discussion states that the pharmacoinvasive strategy 'may have an important reduction on mortality,' yet the certainty of evidence for this outcome is classified as low. Could the authors clarify how they reconcile this conclusion with the statistical uncertainty in the mortality estimate?

9) The time from fibrinolysis to PCI varied significantly across studies (135 minutes to 16.7 hours), which could influence clinical outcomes. Considering that timely reperfusion is a key determinant of efficacy, did the authors assess whether this variability had any impact on treatment effect? If no formal stratified analysis was performed, could the authors clarify whether exploratory approaches (e.g., narrative assessment of trends across studies) were considered to address this clinically relevant issue?

10) The proportion of male participants varied widely across studies (20.4% to 85.9%), and prior research (e.g., the TRANSFER-AMI trial) has suggested potential gender-based differences in treatment response. Given this known variability, did the authors consider any assessment—either quantitative or qualitative—of how gender distribution might have influenced the results? If not, could the authors clarify how this potential confounder was accounted for in the overall interpretation?

6. PLOS authors have the option to publish the peer review history of their article (what does this mean? ). If published, this will include your full peer review and any attached files.

**Do you want your identity to be public for this peer review?** For information about this choice, including consent withdrawal, please see our Privacy Policy .

Reviewer #1: No

Reviewer #2: No

---

## [Author Response · Author response to Decision Letter 1]

22 Apr 2025

Reviewer Nº1:

1) On page 10, line 51-53 the authors state:

STEMI is a public health problem in both industrialized and developing countries with a prevalence that can reach up to 40%.

I am not sure what the authors mean here. This suggests that 40% of the population are having a STEMI at any one time. This needs to be corrected.

Response: We appreciate the comment. Our intention was to emphasize the high burden of STEMI within acute coronary syndromes, so we have modified the text on lines 52 to 54 with the following: " STEMI is a public health problem in both industrialized and developing countries with an estimated global prevalence of over 3 million cases per year, making it the predominant acute manifestation of coronary heart disease”.

2) The repeated use of the word important for statistically significant differences between treatment groups is unnecessary and should be changed.

Response: We appreciate this observation. Following GRADE guidelines 26 and 32, the communication of results is based on absolute point effect and certainty of evidence. Because it is a minimally contextualized approach, the word important is used.

https://pubmed.ncbi.nlm.nih.gov/33857619/

https://pubmed.ncbi.nlm.nih.gov/31711912/

3) Where a statistically significant difference has not been identified stating that there may be an important difference between groups does not mean anything. It might be discussed that a larger patient population would be required to identify a difference and that there is a possibility of type II error. This requires correction in the results and discussion section.

Response: Our phrasing follows GRADE 26 guidelines, which recommend basing conclusions on the absolute effect and certainty of the evidence rather than statistical significance. The interpretation of our results is aligned with these principles, and we are not considering type II error, as GRADE does not prioritize statistical significance in drawing conclusions.

4) Some of the forest plots for the important outcomes should be included in the main manuscript and not just in the supplementary appendix.

Response: We agree and will include key forest plots for primary outcomes such as mortality and reinfarction in the main manuscript.

Reviewer Nº2:

1) While the review focuses exclusively on randomized controlled trials, observational studies could provide valuable real-world insights, particularly for safety outcomes such as bleeding and stroke. Could the authors clarify whether observational studies were considered at any stage of the study selection process?

Response: Our focus was on RCTs due to their methodological robustness. However, we recognize that observational studies may offer valuable data on rare adverse events. To clarify this, we have added in lines 307- 309: “We excluded observational studies to focus on RCTs, which minimize bias and provide the most reliable evidence on efficacy and safety. However, we acknowledge that this may limit the detection of less frequent adverse events.”

2) The search strategy included PubMed, Web of Science, Embase, and Cochrane Library (CENTRAL), but no justification was provided for excluding other potentially relevant databases such as Scopus, LILACS, or CINAHL. Given that broader searches could reduce publication bias and increase comprehensiveness, could the authors clarify the rationale for limiting the search to these four databases?

Response: We appreciate this observation. Our database selection followed Cochrane's recommendations, which suggest prioritizing Medline (PubMed), Central and Embase, we also included ClinicalTrials.gov, and references from included studies. These databases provide comprehensive coverage of clinical trials.

3) The manuscript states that publication bias could not be statistically assessed due to the small number of studies (7 RCTs, 3,053 patients). However, the absence of a formal statistical assessment does not preclude a qualitative evaluation. Did the authors conduct any qualitative assessment of publication bias (e.g., reviewing asymmetries in reported outcomes, searching for unpublished trials, or assessing selective reporting)?

Response: We acknowledge the importance of evaluating publication bias. A common indicator of this bias is the presence of small studies with positive results and funded by the industry. However, we did not find such characteristics among the included studies, suggesting that publication bias is unlikely to be a significant issue in our analysis. We added the following to lines 129–130: " The publication bias statistical assessment could not be performed due to the small number of studies per outcome, but there was no qualitative suspicion of publication bias."

4) Figure 2 illustrates that some studies presented a high risk of bias, particularly in blinding of participants and allocation concealment. However, there is no explicit mention of whether studies with high risk of bias were automatically excluded or included in the meta-analysis without restrictions. The Discussion states that most studies had a low risk of bias, but it does not clarify how studies with high risk were handled. Were these studies included in the main analysis? If so, was any sensitivity analysis performed to assess their impact on the overall results?

Response: We conducted a sensitivity analysis excluding studies at high risk of bias. We added to the manuscript in lines 267–271: “In the sensitivity analysis excluding studies at high risk of bias, the results remained consistent with the primary analysis. The benefit of the intervention was more pronounced for outcomes such as mortality, myocardial infarction, and cardiogenic shock, though the overall conclusions were unchanged. For major bleeding, stroke, and angina, effect estimates were comparable to those observed in the main analysis.”

Corresponding sensitivity analysis forest plots have been included in the supplementary material.

5) The study considers a trial as 'low risk of bias' if at least 5 out of 7 domains in the Cochrane RoB 1.0 tool are classified as low risk. However, Cochrane guidelines do not recommend a fixed numerical threshold for determining overall risk, as certain domains (e.g., allocation concealment or outcome blinding) can have a disproportionately high impact on validity. Could the authors clarify why this specific threshold was chosen instead of a more qualitative domain-based assessment?

Response: Thanks for the suggestion. We used the Cochrane Risk of Bias 1.0 tool, which lacks explicit guidance for general risk classification. This criterion was based on previous systematic reviews, such as the following:

https://www.thelancet.com/journals/lancet/article/PIIS0140-6736(24)01357-6/fulltext#app-1

6) The manuscript defines I² <40% as 'not important heterogeneity.' However, according to Cochrane guidelines, an I² between 30-60% can indicate moderate heterogeneity. Could the authors clarify the rationale behind this threshold? Was this classification based on an alternative reference, such as the GRADE Working Group or specific cardiovascular studies? Additionally, some outcomes with I² >40% (e.g., long-term angina, I² = 76%) were not downgraded in certainty. Could the authors clarify why heterogeneity was not consistently considered when assessing the certainty of evidence?

Response: We revised our interpretation to align with Cochrane's classification, acknowledging that 0 - 40% represents “might not be important”.

https://training.cochrane.org/handbook/current/chapter-10

7) The 'Certainty of Evidence' section states that imprecision was assessed according to GRADE guidelines, considering confidence intervals crossing MID thresholds and the number of events. While Table 2 presents the MID values used, it does not specify how they were determined or applied in the downgrading process. Could the authors clarify whether these MID thresholds were based on predefined literature standards, expert consensus, or empirical data from the included studies? Additionally, what specific criteria led to downgrading by one or two levels for each outcome?

Response: MID thresholds were based on literature and previous studies in cardiology. The article defining the MID thresholds is already cited in the review.

8) In Table 2, the certainty of evidence for 30-day mortality was downgraded by two levels due to very serious imprecision. However, the reported risk ratio (RR 0.87, 95% CI: 0.59–1.27) suggests a potentially beneficial effect, though the confidence interval includes 1. Other outcomes with similarly wide confidence intervals were downgraded only one level. Could the authors clarify the rationale for downgrading mortality by two levels instead of one? Additionally, the Discussion states that the pharmacoinvasive strategy 'may have an important reduction on mortality,' yet the certainty of evidence for this outcome is classified as low. Could the authors clarify how they reconcile this conclusion with the statistical uncertainty in the mortality estimate?

Response: The confidence interval includes both boundaries of the minimal important difference (MID), indicating high uncertainty regarding the true effect. This justifies downgrading the certainty of evidence by two levels.

GRADE 32: https://www.jclinepi.com/article/S0895-4356(21)00108-6/fulltext

9) The time from fibrinolysis to PCI varied significantly across studies (135 minutes to 16.7 hours), which could influence clinical outcomes. Considering that timely reperfusion is a key determinant of efficacy, did the authors assess whether this variability had any impact on treatment effect? If no formal stratified analysis was performed, could the authors clarify whether exploratory approaches (e.g., narrative assessment of trends across studies) were considered to address this clinically relevant issue?

Response: We acknowledge this concern. In lines 281 to 283 of the Discussion, we state that a meta-regression analysis could not be performed to assess the influence of treatment efficacy.

10) The proportion of male participants varied widely across studies (20.4% to 85.9%), and prior research (e.g., the TRANSFER-AMI trial) has suggested potential gender-based differences in treatment response. Given this known variability, did the authors consider any assessment—either quantitative or qualitative—of how gender distribution might have influenced the results? If not, could the authors clarify how this potential confounder was accounted for in the overall interpretation?

Response: A meta-regression analysis considering the percentage of male participants would have been ideal, but at least 10 studies are required for such an analysis, and our dataset did not meet this criterion. We did not conduct a subgroup analysis based on gender because all reported subgroup analyses from the included studies are presented in the "Subgroup Analysis" section of the results, no additional subgroups were available in the included studies.

---

## [Decision Letter · Decision Letter 1]

16 Jul 2025

PONE-D-25-03031R1Pharmacoinvasive strategy versus fibrinolytic therapy in adults with ST-elevation myocardial infarction: a systematic review and meta-analysisPLOS ONE

Dear Dr. Goicochea-Lugo,

Thank you for submitting your manuscript to PLOS ONE. After careful consideration, we feel that it has merit but does not fully meet PLOS ONE’s publication criteria as it currently stands. Therefore, we invite you to submit a revised version of the manuscript that addresses the points raised during the review process.

We look forward to receiving your revised manuscript.

Kind regards,

Parisa Fallahtafti

Academic Editor

PLOS ONE

Journal Requirements:

Reviewers' comments:

Reviewer's Responses to Questions

**Comments to the Author**

1. If the authors have adequately addressed your comments raised in a previous round of review and you feel that this manuscript is now acceptable for publication, you may indicate that here to bypass the “Comments to the Author” section, enter your conflict of interest statement in the “Confidential to Editor” section, and submit your "Accept" recommendation.

Reviewer #2: All comments have been addressed

Reviewer #3: (No Response)

2. Is the manuscript technically sound, and do the data support the conclusions?

Reviewer #2: Yes

Reviewer #3: Yes

3. Has the statistical analysis been performed appropriately and rigorously? 

Reviewer #2: Yes

Reviewer #3: No

4. Have the authors made all data underlying the findings in their manuscript fully available?

Reviewer #2: Yes

Reviewer #3: Yes

5. Is the manuscript presented in an intelligible fashion and written in standard English?

Reviewer #2: Yes

Reviewer #3: No

6. Review Comments to the Author

Reviewer #2: After reviewing the authors’ responses, I would like to highlight remaining gaps that have not been adequately addressed and require further clarification.

The authors acknowledged the impossibility of performing a statistical assessment of publication bias due to the limited number of studies, and added in lines 129-130 that no qualitative suspicion of bias was observed. This revision is constructive; however, the qualitative assessment remains limited, as it was based solely on the absence of small industry-funded studies.

The criterion of considering “low risk of bias” as meeting 5 out of 7 domains was justified by referencing previous systematic reviews, with an appropriate citation (p. 15, lines 116-122), thus demonstrating alignment with established methodological precedents. Nonetheless, the authors’ response and the manuscript fail to discuss the relevance of critical domains such as random sequence generation or allocation concealment, whose qualitative weighting could influence the overall risk classification.

The correction to the definition of heterogeneity to align with Cochrane standards (I² 0–40% as “might not be important”) demonstrates responsiveness and adherence to current guidelines. However, the manuscript does not explain why outcomes with substantial heterogeneity – such as long-term angina (I² = 76%, p. 58, lines 241–247) – did not result in a downgrade of the certainty of evidence within the GRADE assessment (Table 2, p. 56).

The justification of MIDs was based on the literature, with citation of reference [19] (Armstrong et al., 2013; p. 65, line 137), a clinical trial comparing fibrinolysis and primary PCI, in an effort to align with GRADE guidance. However, reference [19] does not explicitly define MIDs or describe the process for determining them, and neither the response to the reviewer nor the manuscript (p. 16, lines 142–144) provides sufficient detail on how MID values were selected or derived. While data from [19] (e.g., a 1.9% difference in the composite outcome) may have been used to infer MID thresholds, the lack of transparency regarding this derivation process compromises methodological clarity.

Reviewer #3: Thank you dear editor for the opportunity to review this systematic review and meta-analysis comparing pharmacoinvasive strategy (PS) with fibrinolysis in ST-elevation myocardial infarction (STEMI). The topic is clinically relevant, particularly in resource-limited settings.

1. Title & Abstract

Title

The title implies a direct comparison between PS and fibrinolysis, but the PS intervention includes fibrinolysis followed by PCI. This could mislead readers into thinking PS excludes fibrinolysis (Page 1, Title).

Abstract

No mention of registration (PROSPERO) or risk of bias assessment tool (Page 1, Abstract).

Absolute effects lack context without baseline risks (Page 1, Lines 35–36).

The statement "PS has trivial or no effect on major bleeding" (Line 41) is overstated; the CI (-17 to +13) includes both trivial harm and benefit, making "no effect" speculative.

The abstract claims "high certainty" for reinfarction but omits downgrading reasons for other outcomes (Page 1, Lines 35–46).

Conclusion: States PS "may have an important reduction of mortality and stroke" but does not clarify that CIs include null effects (Mortality 95% CI: -16 to +10; Stroke: -10 to +2) (Page 1, Lines 44–46).

2. Introduction

Fails to justify why PS (fibrinolysis plus PCI) is compared to fibrinolysis alone instead of primary PCI, the gold standard (Page 13, Lines 66–72).

The response to Reviewer #1 (Page 2, Lines 1–3) revised STEMI prevalence to "over 3 million cases per year," yet the manuscript retains "prevalence that can reach up to 40%" (Page 46, Line 54).

The objective ("to compare PS versus fibrinolysis") is vague; it should specify efficacy/safety outcomes (Page 14, Line 77).

3. Methods

Eligibility Criteria

"Primary PCI performed 2–24 hours after fibrinolysis" (Page 14, Line 85) is ambiguous. Does "primary PCI" imply immediate PCI, conflicting with the 2–24h window?

"Cardiac failure," "angina," and "stroke" lack standardized diagnostic criteria (Page 14, Lines 88–92).

Search Strategy

Database Justification: No rationale for excluding Scopus/LILACS/CINAHL despite Reviewer #2's query (Page 15, Lines 96–99; Page 2, Response #2).

Study Selection & Data Extraction

The "diriment author" (Page 15, Line 105) role is not defined.

Risk of Bias (RoB)

Using a fixed threshold (5/7 low-risk domains = "low RoB") contradicts Cochrane guidelines, which discourage numerical thresholds (Page 15, Lines 121–122; Page 3, Response #5).

Figure 2 (Page 35) uses "+" and "○" without a legend.

Statistical Analysis

Defining I² <40% as "not important" conflicts with Cochrane (30–60% = moderate heterogeneity) (Page 49, Line 127; Page 3, Response #6).

Minimal important differences (MIDs) for outcomes (mortality: ±2/1000) lack justification. No citation or method for deriving MIDs is provided (Page 22, Lines 192–194; Page 3, Response #7).

Qualitative assessment ("no suspicion of bias") is inadequate without funnel plots or registry searches (Page 49, Lines 128–129).

Subgroup Analysis

Pre-specified subgroups ("bled at time of fibrinolysis") are not supported by included studies, yet this is not acknowledged (Page 50, Lines 134–135).

4. Results

Study Characteristics

Table 1 Errors:

"Time from fibrinolysis to PCI" for GRACIA-1 is listed as "16.7 hours (5.6)" without units (Page 20).

Rescue PCI rates in the fibrinolysis group (14–34.9%) are reported but not contextualized (Page 17, Line 180).

Risk of Bias

4/7 studies had high RoB in ≥1 domain, but sensitivity analysis (Page 26, Lines 257–261) was limited to supplemental material, not main text.

Meta-Analysis Results

Revascularization outcomes (30-day/longest) are labeled "high certainty" despite being based on only 2–3 studies (Page 22, Table 2).

Angina at longest follow-up had I²=76% but was not downgraded for inconsistency (Page 25, Line 251).

Mortality is downgraded twice for imprecision (RR 0.87, 95% CI: 0.59–1.27), but the Discussion claims PS "may have an important reduction" (Page 23, Table 2; Page 27, Line 271).

5. Discussion

Highlights PS benefits but downplays null findings (major bleeding: RR 0.91, 95% CI: 0.66–1.26) (Page 27, Lines 276–278).

Studies were from high-income countries (Canada, Europe), but implications are extended to LMICs without caveats (Page 28, Line 285).

Limitations

Time-to-PCI Variability: The impact of widely varying fibrinolysis-to-PCI times (135 min–16.7 hours) on outcomes is not discussed (Page 28, Lines 293–295).

Despite known gender disparities in STEMI, subgroup analysis was not performed due to "insufficient studies," but TRANSFER-AMI reported gender-specific data (Page 26, Lines 263–268; Page 4, Response #10).

6. Conclusion

Claims PS reduces mortality and stroke despite low certainty and CIs including null effects (Page 30, Lines 348–351).

Does not address uncertain evidence for cardiac failure/cardiogenic shock (very low certainty).

7. PLOS authors have the option to publish the peer review history of their article (what does this mean? ). If published, this will include your full peer review and any attached files.

**Do you want your identity to be public for this peer review?** For information about this choice, including consent withdrawal, please see our Privacy Policy .

Reviewer #2: No

Reviewer #3: No

While revising your submission, please upload your figure files to the Preflight Analysis and Conversion Engine (PACE) digital diagnostic tool, https://pacev2.apexcovantage.com/ . PACE helps ensure that figures meet PLOS requirements. To use PACE, you must first register as a user. Registration is free. Then, login and navigate to the UPLOAD tab, where you will find detailed instructions on how to use the tool. If you encounter any issues or have any questions when using PACE, please email PLOS at figures@plos.org. Please note that Supporting Information files do not need this step.

---

## [Author Response · Author response to Decision Letter 2]

1 Sep 2025

Reviewer #2: After reviewing the authors’ responses, I would like to highlight remaining gaps that have not been adequately addressed and require further clarification.

The authors acknowledged the impossibility of performing a statistical assessment of publication bias due to the limited number of studies, and added in lines 129-130 that no qualitative suspicion of bias was observed. This revision is constructive; however, the qualitative assessment remains limited, as it was based solely on the absence of small industry-funded studies.

response: We appreciate the observation. According to GRADE Guidelines 5, “it is very difficult to be confident that publication bias is absent, and almost equally difficult to know where to place the threshold and rate down for its likely presence. Recognizing these challenges, the term GRADE suggests using in evidence profiles for publication bias is ‘undetected.’” Following this recommendation, we have modified the corresponding paragraph in the manuscript to: “The publication bias statistical assessment could not be performed due to the small number of studies per outcome, but publication bias was undetected based on a qualitative evaluation of small trials with positive results and industry-funded studies.”

The criterion of considering “low risk of bias” as meeting 5 out of 7 domains was justified by referencing previous systematic reviews, with an appropriate citation (p. 15, lines 116-122), thus demonstrating alignment with established methodological precedents. Nonetheless, the authors’ response and the manuscript fail to discuss the relevance of critical domains such as random sequence generation or allocation concealment, whose qualitative weighting could influence the overall risk classification.

response: We agree with the suggestion and additionally considered the qualitative implications of the risk of bias according to the importance of its items.

We clarified this in the Methods: “We considered a study to be at low risk of bias if it had at least 5 low risk items, following criteria used in previous systematic reviews [14]. However, we also considered the qualitative importance of individual domains. Deficiencies in these domains were considered in the interpretation of results, regardless of the overall score.”

and in the Discussion: “Although most domains of the RoB 1.0 tool were judged as low risk in the included studies, several critical aspects should be considered when interpreting our findings. The most frequent limitation was unclear allocation concealment in 5 of the 7 studies, which could lead to selective enrollment of participants, baseline imbalances, and overestimation of the observed treatment effects. In addition, lack of blinding of participants and personnel was observed in 4 of the 7 studies. For subjective outcomes (e.g., angina), this lack of blinding could result in differences in expectations and decision-making by both participants and physicians. However, the critical outcomes in the present study are less prone to subjective influence.”

The correction to the definition of heterogeneity to align with Cochrane standards (I² 0–40% as “might not be important”) demonstrates responsiveness and adherence to current guidelines. However, the manuscript does not explain why outcomes with substantial heterogeneity – such as long-term angina (I² = 76%, p. 58, lines 241–247) – did not result in a downgrade of the certainty of evidence within the GRADE assessment (Table 2, p. 56).

response: We appreciate the observation. In the case of long-term angina (I² = 76%), although the heterogeneity was statistically substantial, the direction of effect was consistent across studies, with all point estimates favoring the pharmacoinvasive strategy. According to GRADE guidance, inconsistency warrants downgrading when variability in results is both substantial and unexplained in a way that could alter the interpretation of the effect. We clarify this aspect in the certainty of the evidence section: “Regarding heterogeneity, we rated down one level if I2 was between 40-80% with inconsistency of the direction of the association and rated down two levels if I2 was >80% with inconsistency of the direction of the association.”

The justification of MIDs was based on the literature, with citation of reference [19] (Armstrong et al., 2013; p. 65, line 137), a clinical trial comparing fibrinolysis and primary PCI, in an effort to align with GRADE guidance. However, reference [19] does not explicitly define MIDs or describe the process for determining them, and neither the response to the reviewer nor the manuscript (p. 16, lines 142–144) provides sufficient detail on how MID values were selected or derived. While data from [19] (e.g., a 1.9% difference in the composite outcome) may have been used to infer MID thresholds, the lack of transparency regarding this derivation process compromises methodological clarity.

response: We appreciate the observation, we have detailed the process we followed to establish MIDs in Methods: “To assess imprecision, we followed the GRADE guidelines 34, which recommend establishing minimally important differences (MID) for each outcome.[19] To define the MIDs, we first searched PubMed for studies that specified these values for the target population; however, no such studies were identified. The search strategy is detailed in Supplementary Table 3. Consequently, we determined the MIDs for each outcome based on the results of a trial that compared the most effective intervention for STEMI patients (primary PCI) with fibrinolysis alone (the comparator group in the present study). [20] For outcomes not specified in that trial, the MIDs were determined by consensus among the authors of the present study.”

Reviewer #3: Thank you dear editor for the opportunity to review this systematic review and meta-analysis comparing pharmacoinvasive strategy (PS) with fibrinolysis in ST-elevation myocardial infarction (STEMI). The topic is clinically relevant, particularly in resource-limited settings.

1. Title & Abstract

Title

The title implies a direct comparison between PS and fibrinolysis, but the PS intervention includes fibrinolysis followed by PCI. This could mislead readers into thinking PS excludes fibrinolysis (Page 1, Title).

response: To avoid ambiguity, we have revised the title to clarify that the comparator is fibrinolysis alone. The title now reads: “Pharmacoinvasive strategy versus fibrinolytic therapy alone in adults with ST-elevation myocardial infarction: a systematic review and meta-analysis.”

Abstract

No mention of registration (PROSPERO) or risk of bias assessment tool (Page 1, Abstract).

response: We agree with this observation, and we modified the Methods section of the abstract to read as follows:

“The review protocol was registered in PROSPERO (CRD42022309130). We included randomized controlled trials (RCTs), assessed risk of bias with the Cochrane Risk of Bias 1.0 tool, and calculated pooled relative risks and mean differences.”

Absolute effects lack context without baseline risks (Page 1, Lines 35–36).

response: We have added in abstract the absolute effects together with baseline risks for context

The statement "PS has trivial or no effect on major bleeding" (Line 41) is overstated; the CI (-17 to +13) includes both trivial harm and benefit, making "no effect" speculative.

response: According to GRADE, conclusions should not rely solely on statistical significance, as this reflects only one aspect of the certainty of evidence: imprecision. The minimally contextualized GRADE approach focuses on whether the estimated effect is minimally important, based on the position of the point estimate in relation to the MID thresholds. The final phrasing of results depends on both the location of the point estimate and the certainty of evidence (high, moderate, low, or very low). This approach is described in GRADE guideline 26 and, more specifically for systematic reviews with minimally contextualized approach, in guideline 34. For the outcome of revascularization, considering a MID of ±17, the 95% confidence interval (-17 to +13) marginally overlaps the lower threshold. Therefore, we decided not to downgrade the certainty for imprecision.

The abstract claims "high certainty" for reinfarction but omits downgrading reasons for other outcomes (Page 1, Lines 35–46).

response: We appreciate this observation. Due to the journal’s word limit for the abstract (300 words), it was not possible to include all downgrading reasons for each outcome. The full GRADE assessment, including downgrading reasons for all other outcomes is detailed in the results section.

Conclusion: States PS "may have an important reduction of mortality and stroke" but does not clarify that CIs include null effects (Mortality 95% CI: -16 to +10; Stroke: -10 to +2) (Page 1, Lines 44–46).

response: According to GRADE, conclusions should not rely solely on statistical significance, as this reflects only one aspect of the certainty of evidence: imprecision. The minimally contextualized GRADE approach focuses on whether the estimated effect is minimally important, based on the position of the point estimate in relation to the MID thresholds. The final phrasing of results depends on both the location of the point estimate and the certainty of evidence (high, moderate, low, or very low). This approach is described in GRADE guideline 26 and, more specifically for systematic reviews with minimally contextualized approach, in guideline 34.

2. Introduction

Fails to justify why PS (fibrinolysis plus PCI) is compared to fibrinolysis alone instead of primary PCI, the gold standard (Page 13, Lines 66–72).

response: We agree with your observation, we have modified the introduction: “As a result, fibrinolysis alone remains the most commonly available and feasible reperfusion strategy in these settings. In this context, the pharmacoinvasive strategy combines early fibrinolysis followed by routine coronary angiography and potential PCI within 2 to 24 hours and emerges as a practical and guideline-endorsed alternative when timely PCI cannot be performed [5,8,9]. Importantly, while primary PCI is the gold standard of treatment, it is important to compare the pharmacoinvasive strategy with fibrinolysis alone because this reflects a more realistic and context-appropriate clinical decision pathway in resource-limited environments, where the majority of patients do not have access to timely primary PCI [10,11]”

The response to Reviewer #1 (Page 2, Lines 1–3) revised STEMI prevalence to "over 3 million cases per year," yet the manuscript retains "prevalence that can reach up to 40%" (Page 46, Line 54).

response: The phrase “prevalence that can reach up to 40%” has been removed to avoid confusion. The manuscript now states: “STEMI is a public health problem… with an estimated global prevalence of over 3 million cases per year.”

The objective ("to compare PS versus fibrinolysis") is vague; it should specify efficacy/safety outcomes (Page 14, Line 77).

response:We agree. The objective has been revised to specify the efficacy and safety outcomes assessed: " the objective of this systematic review (SR) is to assess the efficacy and safety of pharmacoinvasive strategy compared with fibrinolysis in adults with ST-segment elevation myocardial infarction. The outcomes assessed included mortality, reinfarction, recurrent ischemia, need for revascularization, length of hospital stay, stroke, major bleeding, heart failure, and cardiogenic shock."

3. Methods

Eligibility Criteria

"Primary PCI performed 2–24 hours after fibrinolysis" (Page 14, Line 85) is ambiguous. Does "primary PCI" imply immediate PCI, conflicting with the 2–24h window?

response: The term “primary” has been removed. The sentence now reads: “PCI performed over a period of time of 2 to 24 hours from the end of fibrinolysis administration”.

"Cardiac failure," "angina," and "stroke" lack standardized diagnostic criteria (Page 14, Lines 88–92).

response: We have revised and clarified the definitions of the specified outcomes to ensure consistency with the terminology and criteria used in the included studies. Specifically, we changed the term "angina" to "recurrent ischemia", which better reflects the definitions applied across trials. The definitions of heart failure and stroke remained consistent and required no modification. Below, we provide a summary of how these outcomes were defined in the included studies:

Heart failure

Armstrong 2006: (3) Congestive heart failure: (i) Physician’s decision to treat CHF with a diuretic, intrave nous inotropic agent or intravenous vasodilator and either (a) the presence of pulmonary edema or pulmonary vascu lar congestion on chest X-ray believed to be of cardiac cause or (b) at least two of the following: (1) rales greater than one-third up the lung fields believed to be due to CHF. (2) PCWP .18mmHg (3) Dyspnoea, with documented pO2 less than 80 mmHg on room air or O2 saturation ,90% on room air, without significant lung disease.

Cantor 2009: heart failure that required treatment 6 hours or more after enrollment and either pulmonary ede ma on a chest radiograph, rales, or a pulmonary capillary wedge pressure greater than 18 mm Hg.

Angina

Bohmer 2010: unstable angina (chest pain at rest suspicious for coronary disease with or without ECG changes), recurrent angina grade II to IV (Canadian Cardiovascular Society classification)

Cantor 2009: Recurrent ischemia was defined as chest pain lasting 5 minutes or longer associated with ST-segment or T-wave changes

Mario 2008: recurrent chest pain with ST-segment deviation or defi nite T-wave inversion occurring more than 12 h after randomisation persisting for at least 10 min despite administration of nitrates, β blockers, or calcium channel blockers and not fulfilling the diagnosis of myocardial reinfarction.

Scheller 2003: recurrent angina pectoris lasting for more than 15 min despite the administration of nitrates or being accompanied by electrocardiographic changes, pulmonary edema, or hypotension

Fernandez-Aviles 2004: spontaneous (at rest) or stress-induced recurrence of typical angina pectoris (or anginal equivalent) that had to coincide with new ECG abnormalities, or abnormal stress test

Stroke:

Armstrong 2006: Stroke after thrombolysis

Bohmer 2010: new focal, neurological deficit of vascular origin lasting more than 24 h.

Cantor 2009: intracranial hemorrhage

Fernandez-Aviles 2004: intracranial hemorrhage

Mario 2008: intracranial hemorrhage

Scheller 2003: Stroke after thrombolysis

Thiele 2005: fatal stroke or stroke causing significant mental or physical handicap

Search Strategy

Database Justification: No rationale for excluding Scopus/LILACS/CINAHL despite Reviewer #2's query (Page 15, Lines 96–99; Page 2, Response #2).

response: According to the Cochrane Handbook for Systematic Reviews of Interventions, it is recommended that “CENTRAL and MEDLINE, together with Embase (if access to Embase is available to the review team), should be searched for all Cochrane Reviews.” Furthermore, the Handbook also highlights that “searches for studies should be as extensive as possible to reduce the risk of publication bias and to identify as much relevant evidence as possible,” and encourages the inclusion of topic-specific and regional databases, such as CINAHL for nursing-related topics or APA PsycInfo for psychological interventions, as well as regional resources like LILACS.

In line with these recommendations, we conducted a comprehensive search in CENTRAL, MEDLINE, and Embase. Additionally, we included Web of Science, which indexes a broad range of journals, including those from regional databases such as SciELO.

Study Selection & Data Extraction

The "diriment author" (Page 15, Line 105) role is not defined.

Response: We appreciate the reviewer’s observation. We clarified this point in the manuscript by specifying that the “diriment author” acted as a third independent reviewer responsible for adjudicating disagreements.

Risk of Bias (RoB)

Using a fixed threshold (5/7 low-risk domains = "low RoB") contradicts Cochrane guidelines, which discourage numerical thresholds (Page 15, Lines 121–122; Page 3, Respon

---

## [Decision Letter · Decision Letter 2]

25 Sep 2025

Pharmacoinvasive strategy versus fibrinolytic therapy alone in adults with ST-elevation myocardial infarction: a systematic review and meta-analysis

PONE-D-25-03031R2

Dear Dr. Goicochea-Lugo ,

We’re pleased to inform you that your manuscript has been judged scientifically suitable for publication and will be formally accepted for publication once it meets all outstanding technical requirements.

Kind regards,

Parisa Fallahtafti

Academic Editor

PLOS ONE

Reviewers' comments:

Reviewer's Responses to Questions

**Comments to the Author**

1. If the authors have adequately addressed your comments raised in a previous round of review and you feel that this manuscript is now acceptable for publication, you may indicate that here to bypass the “Comments to the Author” section, enter your conflict of interest statement in the “Confidential to Editor” section, and submit your "Accept" recommendation.

Reviewer #2: All comments have been addressed

2. Is the manuscript technically sound, and do the data support the conclusions?

Reviewer #2: Yes

3. Has the statistical analysis been performed appropriately and rigorously? 

Reviewer #2: Yes

4. Have the authors made all data underlying the findings in their manuscript fully available?

Reviewer #2: Yes

5. Is the manuscript presented in an intelligible fashion and written in standard English?

Reviewer #2: Yes

6. Review Comments to the Author

Reviewer #2: All points raised by the reviewers have been addressed, and the necessary clarifications have been incorporated into the manuscript. I am in favor of the manuscript being accepted for publication after this round of revision.

7. PLOS authors have the option to publish the peer review history of their article (what does this mean? ). If published, this will include your full peer review and any attached files.

**Do you want your identity to be public for this peer review?** For information about this choice, including consent withdrawal, please see our Privacy Policy .

Reviewer #2: No

---

## [Editor Report · Acceptance letter]

PONE-D-25-03031R2

PLOS ONE

Dear Dr. Goicochea-Lugo,

I'm pleased to inform you that your manuscript has been deemed suitable for publication in PLOS ONE. Congratulations! Your manuscript is now being handed over to our production team.

Kind regards,

on behalf of

Dr. Parisa Fallahtafti

Academic Editor

PLOS ONE